# The Succession of the Cellulolytic Microbial Community from the Soil during Oat Straw Decomposition

**DOI:** 10.3390/ijms24076342

**Published:** 2023-03-28

**Authors:** Anastasiia K. Kimeklis, Grigory V. Gladkov, Olga V. Orlova, Alexey M. Afonin, Emma S. Gribchenko, Tatiana S. Aksenova, Arina A. Kichko, Alexander G. Pinaev, Evgeny E. Andronov

**Affiliations:** 1All-Russian Research Institute of Agricultural Microbiology, 196608 Saint Petersburg, Russia; 2Department of Applied Ecology, Saint-Petersburg State University, 199034 Saint Petersburg, Russia; 3A.I. Virtanen Institute for Molecular Sciences, University of Eastern Finland, FI-70211 Kuopio, Finland; 4Dokuchaev Soil Science Institute, 119017 Moscow, Russia

**Keywords:** oat straw, chernozem, cellulolytic community, succession, glycoside hydrolases, metagenome sequencing, amplicon sequencing

## Abstract

The process of straw decomposition is dynamic and is accompanied by the succession of the microbial decomposing community, which is driven by poorly understood interactions between microorganisms. Soil is a complex ecological niche, and the soil microbiome can serve as a source of potentially active cellulolytic microorganisms. Here, we performed an experiment on the de novo colonization of oat straw by the soil microbial community by placing nylon bags with sterilized oat straw in the pots filled with chernozem soil and incubating them for 6 months. The aim was to investigate the changes in decomposer microbiota during this process using conventional sequencing techniques. The bacterial succession during straw decomposition occurred in three phases: the early phase (first month) was characterized by high microbial activity and low diversity, the middle phase (second to third month) was characterized by low activity and low diversity, and the late phase (fourth to sixth months) was characterized by low activity and high diversity. Analysis of amplicon sequencing data revealed three groups of co-changing phylotypes corresponding to these phases. The early active phase was abundant in the cellulolytic members from Pseudomonadota, Bacteroidota, Bacillota, and Actinobacteriota for bacteria and Ascomycota for fungi, and most of the primary phylotypes were gone by the end of the phase. The second intermediate phase was marked by the set of phylotypes from the same phyla persisting in the community. In the mature community of the late phase, apart from the core phylotypes, non-cellulolytic members from Bdellovibrionota, Myxococcota, Chloroflexota, and Thermoproteota appeared. Full metagenome sequencing of the microbial community from the end of the middle phase confirmed that major bacterial and fungal members of this consortium had genes of glycoside hydrolases (GH) connected to cellulose and chitin degradation. The real-time analysis of the selection of these genes showed that their representation varied between phases, and this occurred under the influence of the host, and not the GH family factor. Our findings demonstrate that soil microbial community may act as an efficient source of cellulolytic microorganisms and that colonization of the cellulolytic substrate occurs in several phases, each characterized by its own taxonomic and functional profile.

## 1. Introduction

In agriculture, the production of grain is accompanied by the production of straw, whose yield surpasses the target product [1,2,3]. There are ways of handling excessive straw quantities, differing in their economic and labor costs. One of the most cost-effective ways of utilizing excessive straw is burning, but it wastes potentially valuable resources and results in severe environmental consequences, including gas emissions and the negative impact of heat on soil fertility [2,4]. Other ways of straw usage include biofuel production [5], the investigation of which is a promising research direction. However, it requires straw transportation, which induces extra costs. So, processing straw at the origin site can be a solution to multiple problems. The reintroduction of straw into the field solves both the problems of transportation costs and nutrient loss. It prevents soil erosion and involves plant residues in the global carbon cycle [6,7]. However, this method has some disadvantages to overcome. Straw provides some easily digestible carbohydrates, proteins, lipids, and minerals, but it mostly consists of recalcitrant lignocellulose. Additionally, the introduction of bare straw into the soil shifts the ratio of carbon to nitrogen, which must be compensated for its effective assimilation by microorganisms. Thus, the search for ways of the effective processing of straw is still an acute problem for agriculture.

Since straw is a complex raw substrate, its decomposition requires the work of multiple enzyme systems, found in a variety of bacteria and fungi. Cellulose, as the main component of straw, is decomposed by enzymes, most of which are listed in the Carbohydrate-Active EnZymes database (CAZy) [8]. The biggest class of enzymes in CAZy comprises Glycoside hydrolases (GH), which are currently divided into 173 families based on the amino acid sequence similarity [9]. GH class encompasses enzymes, aimed at the glycosidic bond between carbohydrates or a carbohydrate and a non-carbohydrate moiety [10]. Consequently, cellulose decomposition is carried out by multiple, but not all, enzyme families across the GH classes. Different families include enzymes aimed mainly at the β-1,4 links in the polysaccharide chain of the recalcitrant cellulose (β-glucosidases, exo-β-glucanases, and endo-β-glucanases) and hemicellulose molecules (β-xylosidase, β-mannanase; β-mannosidase, β-xylanase, etc.), gradually breaking ithem into more accessible compounds [11]. The main families containing these enzymes are GH1, GH3, GH5, GH6, GH7, GH9, GH10, GH30, GH43, and others [11,12]. In natural habitats, these enzyme systems are distributed among different members of the microbial community [13]. Understanding the principles of formation and functioning of the cellulolytic microbial consortium is essential knowledge for the formulation of highly effective preparations for straw decomposition.

Soil from different environments can serve as a source of cellulolytic microorganisms. A number of studies focused on the isolation of single strains from various soil types [14,15,16,17,18], but this approach has several flaws. It was reported that cellulolytic bacteria may take up to a fifth of the total soil community [19]. Additionally, it has been shown that many families of enzymes are simultaneously involved in the decomposition of straw, and different functions are distributed between different members of the microbial community, making it impossible to isolate a single “most important” member [20]. So, the complex task of straw degradation is achieved by the association of microorganisms acting together. Thereby, there is still an ongoing search for cellulolytic microbial consortia which would facilitate straw decomposition.

Multiple studies have shown that during the composting of untreated straw with a natural epiphytic microbiome, the microbial community undergoes taxonomic and functional succession [21,22]. Meanwhile, straw introduction into the soil creates a surplus of nutrients, specifically carbon compounds, which facilitates a new path in the microbiota succession [23]. The aim of this study was to grow a de novo cellulolytic community on sterile straw using soil as a source of degrading microorganisms and exploring its succession stages. As a source of microbiota, we chose chernozem, a soil type common in the southern regions of Russia. Cellulolytic capabilities of the chernozem microbiome were reported earlier [24,25]. Our team has already worked with chernozem and demonstrated that it can be a potential source of cellulolytic microorganisms by both traditional microbiology and molecular methods [26,27]. As a source of a lignocellulolytic substrate, we chose oat (*Avena*), a widely cultivated forage crop. A model laboratory experiment of colonizing sterile straw by soil microbiota was set up in order to study the succession of the oat straw decomposition community. We analyzed microbial activity by the measurement of soil respiration (SR), taxonomy succession by the sequencing of 16S rRNA gene for prokaryotes and ITS2 region for fungi on the Illumina Miseq platform, cellulolytic potential of the resulting community by the search for GH genes in the nontargeted metagenome obtained on the Oxford Nanopore MinION platform and functional succession using real-time PCR of the GH gene selection.

## 2. Results

### 2.1. Microbial Activity

During the 6 months of the experiment, notable decomposition of straw in nylon sachets was observed. Maximum SR values were detected at the beginning of the experiment, and they declined towards the end. According to one-way ANOVA, carbon dioxide emission rates were separated into three groups with significantly different SR values (*p*-value ≤ 0.05), from high to low: (a) 3–21 days, (b) 28–35 days, and (c) 42–182 (Figure 1). The SR experimental values in the first two groups were significantly higher than controls. In the last group, SR experimental values were higher than controls until 133 days, though not significantly in all measurements except one. According to the dynamics of carbon dioxide emission, three phases of microbial activity were distinguished: (1) early, which lasted for the first month; (2) middle, which lasted until the third month; and (3) late, which lasted until the end of the experiment. In the early phase, activity was the highest and was rapidly decreasing towards the end. In the middle phase activity continued to decrease but at a slower pace. In the late phase, stabilization of activity occurred.

At the end of each phase, procaryotic and fungal quantities were assessed by calculating ribosomal operons per 1 g of the substrate. It showed a significant increase of bacterial ribosomal operon from the first to the latter phases (*p*-value = 0.00421 and 0.000183 respectively) (Appendix A). Fungal ribosomal operon numbers decreased between phases but not significantly.

In accordance with the results of the SR measurement, subsequent taxonomical analysis of the dynamics of microbial colonization of straw was performed on substrates from ten sampling periods, covering the entire experiment and different phases of microbial activity: early (days 3, 14, 28), middle (days 49, 63, 91), and late (days 119, 140, 161, 182).

### 2.2. Microbial Diversity

In total, 41 out of 42 libraries of 16S rRNA gene amplicons were left after a quality check. Data from all libraries amounted to 624,236 reads with a median of 13,850, which were attributed to 2062 phylotypes (Appendix A). For the ITS2 fragment amplicons, all 42 libraries passed a quality check. In total, 460,040 reads were acquired, with a median of 8,278.5. Data were attributed to 3,178 phylotypes, but only 43% were assigned to a known kingdom. The decomposing community differed from bulk soil microbiome; they had only 102 common phylotypes of bacteria (22.4% of reads) and 95 common phylotypes of fungi (42.2% of reads) (Appendix A).

Both bacterial richness and evenness of the straw decomposing community, assessed by three alpha diversity indices (Observed, Shannon, and Inverted Simpson), significantly increased during the experiment (Figure 2a). The lowest values were detected on day 3, which was the earliest sampling point in the analysis; the highest values were reached on day 119, which marks the beginning of the late phase (Appendix A). Alpha diversity indices were negatively correlated with SR values, as shown by Pearson’s product-moment coefficient (−0.6980158, *p*-value = 0.02479) (Figure 2c). Divided into phases, the alpha diversity indices of samples from the early and middle phases did not differ significantly from each other but were significantly lower than those from the last phase. At the same time, the measurement of MPD (mean pairwise distances) showed that the early phase was significantly less diverse than the later phases (*p*-value ≤ 0.001) (Appendix A). The early phase was marked by increasing microbial diversity. In the middle phase, the increase slowed down. In the late phase, diversity abruptly reached its maximum values and stayed stable until the end of the experiment. Alpha-diversity of decomposing community remained lower than diversity of the control chernozem soil during all phases (Appendix A).

Beta diversity of bacterial community marked differences between different stages of straw colonization, which coincided with alpha diversity. According to PERMANOVA, the dispersion of samples was higher between microbial communities of different phases than within (F = 8.2033, *p*-value ≤ 0.001). Bacterial samples of the decomposition experiment and control soil were separated along the *X*-axis of the NMDS plot, while samples from different phases of decomposition were separated along the *Y*-axis (Figure 3a). Dynamics of the decomposing microbiota samples were more pronounced in the early phases than in the latter. Stepwise comparison of beta diversity between the earliest sample with the following ones showed an acceleration of dynamics in the early phase, then a slowdown in the middle with an abrupt increase before the last phase (Appendix A).

For the eukaryotic part of the straw-decomposing community, no such tendencies were revealed as they were for the bacterial part. The evenness and richness of the fungi, according to the alpha diversity indices, did not differ significantly between samples and no phases could be distinguished (Figure 2b). A similar observation can be made of the beta diversity plot (Figure 3b). NMDS shows shifts in diversity between samples, but it was not unidirectional, as for bacteria. Differences in fungi diversity between bulk soil and experiment samples were not as pronounced as for bacteria.

Thus, according to alpha and beta metrics, the straw-decomposing bacterial community accumulated diversity during the early and middle phases and reached its peak by the fourth month of the experiment, when it could be considered a mature microbial consortium. The fungal part of the community did not show clear dynamics during its succession.

### 2.3. Taxonomy Overview

During prokaryotic succession, the number of represented phyla in the community increased. The first colonizers on the third day were attributed only to four phyla: Pseudomonadota, Bacteroidota, Bacillota, and Actinobacteriota (Figure 4a). On the 14th day Verrucomicrobiota, Myxococcota, Planctomycetota, and Bdellovibrionota appeared. Acidobacteriota appeared on the 49th day. Chloroflexota, Cyanobacterota, Gemmatinomonadota, Spirochaeota, and Thermoproteota appeared on the 91st day. After the 119th day, the maximum presence of bacterial phyla was registered, including Armatimonadota, Ca. Dependentiae, Fibrobacteriota, Nitrospirota, and Patescibacteria. The most frequent genera among bacterial phylotypes were *Chitinophaga*, *Ohtaekwangia*, *Bacillus*, *Rhizobium*, *Pseudomonas*, and *Inquilinus* (Appendix A). Coinciding with the differences, detected by alpha and beta-diversity, taxonomic composition of the decomposing community did not “gravitate” towards microbiome of the control soil but rather developed in its own direction. For example, chernozem soil was abundant in the representatives of Verrucomicrobiota, Acidobacteriota, and Thermoproteota, which did not receive advantage of growing on straw. Therefore, subsequent analysis concentrated on the succession of the decomposing community and not its comparison with the soil microbiome.

Fungal diversity was presented by three phyla during the whole sampling period (Figure 4b). A major part of fungi phylotypes belonged to Ascomycota. Apart from it, there was a presence of Basidiomycota and Mucoromycota representatives on different sampling days. The most frequent fungi phylotypes were attributed to the genus and species level, including *Chloridium aseptatum*, *Lecythophora canina*, *Schizothecium inaequale*, *Albifimbria verrucaria*, and *Conocybe crispa* (Appendix A).

### 2.4. Community Succession

#### 2.4.1. Data Filtering

The peculiarity of the experiment design was that we followed the dynamics of the development of the decomposing community in 10 physically distant compartments–sachets with straw. In order to identify general patterns in the microbiome development and remove random individual outliers of sachets, we left in the analysis only phylotypes found in the decomposing microbiome with the following characteristic: the presence of at least 10 reads in more than 10% of samples. After this filtering, only 321 out of 1063 bacterial phylotypes were left with an additional 101 “major outliers” (Appendix A). For fungi, 68 out of 1264 phylotypes were left in the analysis (Appendix A).

Among bacterial representatives in the individual sachets, some unique phylotypes with high read counts were allocated into the “major outliers” group. Dispersion of these phylotypes between days showed that most of them stood not only as outliers of individual sachets but also as technical replicates within one sachet (Appendix A). Among those were representatives of Pseudomonadota (*Pseudomonas*, *Sphingomonas*, and *Escherichia*), Bacillota (*Fructilactobacillus*, *Levilactobacillus*, and *Lactiplantibacillus*) and Verrucomicrobiota (*Terrimicrobium*) (Appendix A).

The filtered set of universally represented phylotypes was used to access the microbial succession during the phases of straw colonization. Since the diversity of microorganisms increased, it was incorrect to apply pairwise sample comparison methods or compositional data analysis methods to this dataset. Therefore, the WGCNA method after variance stabilizing transformation (DESeq2) was used to formalize the association of bacteria into groups characteristic of different colonization phases. Analysis separated phylotypes into four clusters with distinct patterns (Figure 5a). Three groups coincided with the earlier established division of the experiment into the three phases of microbial activity–early, middle, and late. The fourth group contained phylotypes, universally spread across the experiment.

#### 2.4.2. Bacterial Phases

The first so-called “early” group represented 71 phylotypes, appearing and reaching their maximum in the first month of incubation and disappearing almost completely in later stages. In WGCNA, it corresponds with the salmon cluster (Appendix A). The most abundant phylotypes in this group, which were not necessarily unique in taxonomy for the whole dataset, belonged to Bacteroidota (*Chitinophaga*, *Dyadobacter*, and *Flavobacterium*) and Pseudomonadota (*Cupriavidus*, *Achromobacter*, *Rhizobium*, *Pseudomonas*, and *Lysobacter*). Some of the above and a few more phylotypes from this group were attributed to unique taxa, detected only in this phase, including representatives of Actinobacteriota (*Cellulosimicrobium*, *Glycomyces*, and *Microbacterium*), Bacteroidota (*Chryseobacterium* and *Flavobacterium*), Pseudomonadota (*Achromobacter*, *Neorhizobium*, *Cupriavidus*, *Lysobacter*, *Massilia*, *Ensifer*, *Microvirga*, *Pseudoduganella*, *Stenotrophomonas*, and *Xylophilus*).

The second “middle” phase group represented 29 phylotypes, which reached their maximum by the second month of incubation and persisted in the community onwards. By WGCNA, these phylotypes were assigned to the green cluster (Appendix A). The most prominent representatives belonged to Bacteroidota (*Chitinophaga*, *Ohtaekwangia*), Bacillota (*Bacillus*, *Solibacillus*, *Planococcaceae*, and *Terribacillus*), Pseudomonadota (*Inquilinus*, *Rhizobium*, *Bradyrhizobium*, *Luteibacter*, *Starkeya*, and *Luteimonas*), and Planctomycetota (*Singulisphaera*).

The third most diverse group represented 139 phylotypes, appearing at the late phase, after three months of incubation. These were represented by the red cluster (Appendix A). In this cluster, major representatives belonged to Bacteroidota (*Ohtaekwangia* and Microscillaceae). Numerous representatives of Acidobacteriota, Actinobacteriota (*Conexibacter*, *Galbitalea*, *Dactylosporangium*, *Iamia*, and *Solirubrobacter*), Verrucomicrobiota, Myxococcota, Cyanobacterota, Chloroflexota, Bdellovibrionota, Spirochaeota, Planctomycetota, Thermoproteota, Gemmatimonadota, and others appeared at this stage.

The last group, corresponding to the cyan cluster, contained 82 phylotypes, consistently or without apparent patterns appearing in all samples (Appendix A). Here, most of the universally abundant phylotypes were attributed to *Paenibacillus*, *Starkeya*, *Pseudoflavitalea*, *Niastella*, and *Lysinibacillus*. Sporadic appearance of phylotypes from Bacteroididota (*Ohtaekwangia*, *Chitinophaga*), Actinobacteroidota (*Conexibacter* and *Actinocorallia*), Verrucomicrobiota (*Terrimicrobium*), and others was noted.

To conclude, representatives of Bacteroidota (*Chitinophaga*, *Ohtaekwangia*) were persistent in all phases of bacterial succession, but each phase had its own phylotypes, attributed to these genera. The early phase was characteristic of Gammaproteobacteria representatives, which disappeared later from the community. The middle phase was specific to a wide variety of Bacillota and Alphaproteobacteria, appearing and persisting in the community. The last phase marked the burst of bacterial diversity from different phyla.

#### 2.4.3. Fungal Phases

The WGCNA analysis managed to separate fungi phylotypes into two clusters, one dispersed across all succession (salmon) and one corresponding with the middle-to-late phase (green) (Figure 5b). Coinciding with alpha and beta diversity analyses, many fungal phylotypes were detected at all phases of the experiment, with only some species demonstrating differences according to the day of sampling (Appendix A). *Coprinellus flocculosus* and *Schizothecium inaequale* were appearing in the fungal community since the early phase, while *Chloridium aseptatum*, *Lecythophora canina*, *Marquandomyces marquandii* and *Scytalidium* were appearing mainly after the second month of the experiment. Phylotypes belonging to Ascomycota (*Albifimbria*, Coniochaetaceae, *Gibberella humicola*), Basidiomycota (*Conocybe*, *Occultifur*, *Waitea*), and Mucoromycota (*Actinomucor*) were periodically encountered in the dataset.

### 2.5. Functional Distribution of Glycoside hydrolases in the Mature Decomposing Community

The transition between middle and late phases of the succession of the decomposing community marked maximum microbial diversity. Additionally, SR data showed that after 3 months of the experiment, microbial activity had stabilized. Taking these considerations into account, the 3-month sample, the borderline between the middle and late phases of straw decomposing microbial community succession, was chosen for the functional analysis and the search of the GH genes. The resulting yield of full metagenome sequencing of DNA from the 91-day sample representing this phase was 10.9 Mbp, with N50 of 4886. The metagenome was polished and annotated, and only genes annotated as belonging to the CAZy database were investigated further. The metagenome contained 83.9% bacterial contigs, and only 1.8% belonged to fungi. The rest were attributed to Metazoa, Plants, and Archaea.

According to the CAZy database, the metagenome of the decomposing microbial community contained 1388 GH genes, 1194 of which belonged to Bacteria and 193 to Fungi (Appendix A). As assigned by EggNogg, the most abundant CAZy genes were attributed to Pseudomonadota (Xanthomonadales, Sphingomonadales, Bradyrhizobiaceae, Rhizobiaceae) (455 genes), Bacteroidetes (Sphingobacteriales, Cytophagales) (339 genes), Actinobacteriota (Streptosporangiales) (156 genes), and Bacillota (60 genes) phyla for bacteria and the Ascomycota (Sordariomycetes) (191 genes) phylum for fungi. So, out of the four most major phyla in the bacterial part of the decomposing microbial community detected by Illumina sequencing, all were also represented by the highest quantities of GH genes. However, according to 16S rRNA gene sequencing data, the relative abundance of Bacillota was higher than Actinobacteriota in all analyzed days of the experiment, while the relative content of GH genes attributed to these phyla was reversed.

According to the CAZy classification, the most represented GH families in the metagenome of the three-month-old straw decomposing community were attributed to GH3 (227), GH31 (117), GH18 (114), and GH20 (91). According to the main functions of the presented GH families, three major groups in the metagenome were distinguished: those connected to cellulose degradation (“cellulose” group), those connected to metabolism of simple carbohydrates (“carbohydrate” group), and those connected to chitin degradation (“chitin” group) (Table 1). The main representatives of the “cellulose” group in this dataset belonged to GH3, GH5, GH9, GH30, GH43, and GH94 families. Families from the “carbohydrates” group included GH31, GH95, GH15, and GH77. A notable presence was detected in the families from the “chitin” group, including GH18, GH19, and GH20. All these GH families from all three groups were found in almost all phyla, detected by 16S rRNA and ITS2 amplicon sequencing, and their relative abundance coincided with the taxonomy data. Pseudomonadota had all groups present, and the “cellulose” group was the most abundant, followed by the “carbohydrate” and then the “chitin”. For Bacteroidota, Actinobacteriota, Bacillota, Acidobacteriota, and Planctomycetota phyla “cellulose” and “carbohydrate” groups were equally represented, while the “chitin” group was less present than the other two. The “cellulose” and “carbohydrate” groups were also detected in minor quantities in Verrucomicrobiota, Cyanobacterota, and Chloroflexota. As for the fungal part of the decomposing community, in Ascomycota “the chitin” group of GHs had more matches than the “cellulose” and “carbohydrate” groups. For Basidiomycota, only one gene was found, attributed to the “chitin” group.

To conclude, according to the search of GH genes in the mature straw decomposing microbial consortia, functionally they were represented by GH, involved in cellulose, simple carbohydrates, and chitin utilization. The main carriers of these genes coincided with bacterial and fungal phyla, appearing in the community from the first days of straw colonization.

### 2.6. Succession of GH Genes during Phases of Decomposition

To assess functional dynamics of degradation phases, a set of 23 GH genes, found in the metagenome and connected to cellulose decomposition, was chosen for the primer construction (Appendix A) and real-time PCR analysis. They represented various GH families and were attributed to several genera found in the microbial community by the earlier analyses. The data was log-transformed and difference in phase distribution was calculated relatively to day 3 of the experiment. As a result, most of the tested GH genes showed maximum presence at the middle phase of the cellulose colonization, regardless of their function (Appendix A). The presence of several GH genes did not alter between phases. According to PERMANOVA, differences in the dynamics of the selected GH genes were significantly explained by taxon attribution (R^2^ = 0.54896, *p*-value = 0.007) and not by GH family attribution (R^2^ = 0.18580, *p*-value = 0.499) (Appendix A). This effect is presented on the WPGMA clustering of the real-time data (Figure 6): GH genes are grouped according to the genus and not the GH family. So, in the long-term succession of the microbial community, the presence of the GH genes was determined not by the stage of cellulolytic substrate decomposition but by the microbiota inhabiting the community at a certain point in time.

## 3. Discussion

Soil is a complex substrate containing nutrients in a variety of forms, from easy-to-digest to recalcitrant. Moreover, this environment is under constant biotic and abiotic stress. All this forms a complex soil microbiota, consisting of a plethora of microorganisms adapted to various nutritional and climatic conditions. Earlier studies already used soil as a source of active microbiota in the experiments on the decomposition of various substrates [28,29], but, as a rule, they did not remove the surface microbiome from the substrate, which distorted results of the microbial succession. The setting of our experiment allowed us to exclude this effect and analyze the process of the de novo colonization of the lignocellulosic substrate by the chernozem microbiota and identify its most prominent phases during the long-term experiment.

The chernozem from Kamennaya steppe, which was used in the current experiment, was recognized as having a potentially high biological activity and diversity of microbial communities [30]. We worked with this soil earlier and showed that it contains potential cellulolytic microorganisms, but plating the soil on the cellulose-containing medium drastically shifts contents of the initial microbial community, giving the ecological advantage to those bacteria, which have not previously been predominant [27]. Along with these results, the composition of the mature decomposing community strongly differentiated from the priming soil microbiome, which could be explained by the fact that soil and straw are environmental niches which provide benefits to different groups of microorganisms. The diversity of the cellulolytic community remained lower than those of the primary soil, even after 6 months of incubation. The microbiome of the cellulolytic community carried a resemblance to the soil microbiome, e.g., representatives from Bacillota (*Bacillus*, *Planococcaceae*) and Gammaproteobacteria (*Pseudomonas*, *Massilia*), but in most cases they were not the main components. 

Due to the design of our experiment, the only measured agrophysical parameter was soil respiration (SR), which is defined as the process of carbon dioxide released by microorganisms. The application of this method has shown its effectiveness in assessing microbial activity in response to anthropogenic agricultural practices [31]. Maximum values of SR were detected on the first measurement on the third day of the experiment, after which a significant decline in SR values, specifically after the second month, was detected. Previously the effect of elevated values of SR during cellulose decomposition was associated with the introduction of additional glucose to the substrate [32]. Thus, our results could be explained by the particular substrate of lignocellulose we used: oat straw has a high content of water-soluble carbon, which is more accessible to microorganisms than cellulose [33,34]. It might have led to a higher microbial activity in the early phase connected with the catalysis of simple carbohydrates present in the unaltered straw.

Depending on the design and the duration of the experiment on the straw decomposition, two or three phases could be distinguished in a process of microbial succession [21,22,35]. Our data allowed us to distinguish three phases of bacterial succession during the decomposition of lignocellulosic subtract: early (first month), middle (second to third month), and late (fourth to sixth month). This distinction was supported by the microbial activity assessed by carbon dioxide emission, by the bacterial quantities assessed by the real-time PCR, and by the bacterial dynamics assessed by the high-throughput 16S rRNA gene sequencing. Despite the fact that the experiment was laid in multiple separate nylon bags, the pattern of microbial succession turned out to be common, with the exception of several outlier phylotypes. Each phase was characteristic of a group of microorganisms consisting of several dozens of co-variable bacterial phylotypes. These phylotypes included both taxa unique for each phase and common throughout the experiment. These findings coincide with the functional differences in cellulolytic community between phases: the difference in the patterns of GH gene presence was connected to the bacterial host and not to the family of the enzyme. 

Despite the evidence that the early phase of community formation involved the degradation of simple carbohydrates, early microbial colonizers of straw were potentially cellulose-degrading organisms. Among those appeared representatives of actinomycetes, which are known to be active producers of secondary metabolites [36]. For instance, *Cellulosimicrobium* was reported to be a normal part of soil microbiota [37] and to have cellulase and xylanase activities [38,39,40,41]. Some strains of *Microbacterium* were reported to have cellulolytic activities [42]. However, they reached maximum diversity by the late phase. Some minor representatives from different phyla of the early succession phase, including *Streptomyces* [43], *Chryseobacterium* [44], and *Dyadobacter* [45], were reported to be able to degrade lignocellulose. The early stages were also characterized by a high relative representation of Gammaproteobacteria (*Pseudomonas*, *Cupriavidus*, *Massilia*) and Alphaproteobacteria (Rhizobiaceae); most of them were reported to contain a lot of cellulase-active GHs. It was established by earlier findings that Pseudomonadota, specifically Alpha- and Gammaproteobacteria, play a major role in cellulose decomposition [46]. In accordance with these data, in this study, about half of the GH found in the metagenome of the community involved in the decomposition of cellulose belongs to the representatives of Pseudomonadota.

Bacillota were present at all phases, but most prominently they populated the microbial community in the middle phase. This is consistent with the findings that Bacillota appear after the initial stage of lignocellulose decomposition [47]. Many genera of this phylum, detected in this dataset, were reported to have cellulolytic strains, including *Bacillus* [48,49], *Paenibacillus* [50], *Lysinibacillus* [51]. A relatively low content of GH genes in the representatives of this phylum was shown, but it could be explained by the differences in annotation bases for 16S rRNA and metagenome data and low coverage of the metagenome assembly. 

The most prominent role of the straw decomposition community in this experiment was played by Bacteroidota. A wide range of microorganisms from this phylum is known to play an important role in the decomposition of various polymers [52]. In our work, it was shown that these microorganisms are present at every succession stage, with the succession of some representatives of this phylum (*Chitinofaga*) by others (*Ohtaekwangia*). Moreover, this phylum accounted for the second largest part of GH genes found in the metagenome. It is worth noting that according to the Polysaccharide Utilization Loci (PUL) DataBase, the major representative of the early community *Chitinofaga* is rich in PULs, which is a marker of active cleavage of complex polysaccharide substrates already at the early stages [53]. 

Although a significant proportion of microorganisms not associated with lignocellulose decomposition appear in the later stages of decomposition, we can assume they are an important part of the stable cellulolytic community. For example, it is known that enzymes associated with sulfur metabolism may play an important role in the decomposition of complex straw components, such as polyphenol compounds [54]. The presence of specific nitrifiers and methylotrophs in the community (*Nitrocosmicus*, *Nitrospira*) can play an important role in the construction of efficient communities. The role of nitrogen exchange in catalytic soil systems is underestimated because, in addition to the competition for carbon sources, the high competition for free nitrogen should also be considered [55]. *Starkeya*, one of the major inhabitants of the middle phase, was described to have a chemolithoautotrophic lifestyle, which allows it to both consume carbon dioxide and produce it [56]. *Conexibacter*, which appears in the late phase, was isolated as a soil bacterium, involved in the carbon and nitrogen cycle [57]. The appearance of predatory microorganisms (obligate—Bdellovibrionota, Vampiriovibrionota; facultative—Myxococcota, *Cytophaga*, *Lysobacter*) at the different phases also indicates the reorientation of the community from simple carbohydrate catalysis since it is known to be a powerful factor in the dynamics of microbial succession [58,59]. There is also evidence that some of the genera detected at various phases of decomposition (*Pseudomonas*, *Planctomyces*, *Vampiriovibrio*, *Luteibacter*) can be accompanying microflora, which act as secondary consumers [60]. These examples expand the understanding of the complexity of interactions between community members.

We showed an increase in bacterial diversity and its phylogenetic diversity and succession from a relatively simple cellulolytic community to a complex microbial community of autochthonous microorganisms with variable functions in the community. At the same time, we did not observe an increase in fungal diversity. This may be linked to the difference in the life cycle duration of these groups of microorganisms. Full metagenome sequencing revealed that fungi accounted only for less than 2% of the contigs. It contradicts the real-time data, which showed high quantities of fungal ribosomal operons at the end of the middle phase. This could be due to a number of factors: less efficient fungal DNA isolation, less efficient nanopore sequencing of fungal DNA or lower quality assemblies due to the large size and diversity of fungal genomes.

In contrast to bacterial succession, only two phases were identified in fungal succession, but many major phylotypes were present in all phases. Many fungi found in the community were described as saprophytic with various enzymatic activities. The early phase one was specific in *Schizothecium inaequale*, which is described both as a coprophilous [61] and endophytic fungus [62]. Another endophyte, associated with decaying matter, was *Coprinellus flocculosus*, which is a mushroom-forming fungus [63]. Species associated with the late phase of straw decomposition were reported to be endophytic with high enzymatic activities—*Chloridium aseptatum* [64] and *Scytalidium* [65,66]. Other saprophytic fungi found in the community, usually associated with soil or plants and reported to have high enzymatic activity, were *Albifimbria verrucaria* [67], Chaetomiaceae [68,69], *Occultifur* [70,71], *Waitea circinata* [72]. Coniochaetaceae, which is widely presented in the later phase, is a family with well-known lignocellulolytic fungi [73,74]. The fungal consortium in the early phase was also presented by the known food mold *Actinomucor elegans*, which was reported to have high enzyme activity, including protease, lipase, glutaminase, and others [75,76].

The major phylotypes of fungi described above, with the exception of Coniochaetaceae, are not described in the literature as typical cellulolytic organisms. So, the analysis of the role of the fungal fraction in the current experiment remains unclear. Nevertheless, both bacteria and fungi are important players in lignocellulose decomposition [35,61,77,78]. It is known that, in spite of the fact that the fraction of nucleic acids encoding CAZymes belonging to the bacterial component is relatively superior to the fungal component of the community, functionally it is fungal enzymes that can play the main role in the degradation of the lignocellulosic complex [79,80,81]. It cannot be excluded that the decrease in diversity and the shift of the fungal community from Mucoromycota and Basidiomycota at early stages to Ascomycota at later stages was the result of the antifungal activity of the microbial community. For instance, one of the main components of the core microbial community was *Chitinophaga*, which specializes on mycelium degradation [28]. *Mucilaginibacter*, found in the late phase, potentially can be a mycophagous bacteria [82]. This assumption is also confirmed by the high number of chitinases we found in the bacterial part of the microbial community. The presence of both bacterial and fungal chitinases in the metagenome of the mature decomposing consortium is indicative of the potential counteraction of these two community components.

## 4. Materials and Methods

### 4.1. Experiment Design

The idea of the experiment was to model the dynamics of the formation of cellulolytic consortium from soil microbiota using straw as a substrate and study its colonization process. To achieve this sterilized straw in nylon sachets was submerged in the soil for six months. Fallow chernozem from the Agroecological Station “Kamennaya Steppe” of the Dokuchaev Research Agricultural Institute in the Voronezskaya area was chosen as a source of decomposing microbiota. This soil was removed from crop rotation more than 100 years ago. Before that it was used for sowing wheat. Its characteristics were: C_total_ 4.86 ± 0.12%; pH_salt_ 6.40 ± 0.08; N_total_ 0.533 ± 0.02. As a source of lignocellulose biomass oat (*Avena*) straw was used with the following characteristics: ash 9.98 ± 2.04, N_total_ 1.897 ± 0.012, C:N 23.5, water-soluble carbon 11.8 ± 0.50 g/kg.

The experiment took place in 2018. The soil was ground and sieved at 5 mm, watered to 60% of the full moisture capacity, placed in the 2-liter plastic containers, and left to rest for 2 weeks to eliminate the effect of these manipulations on the CO_2_ emission. Straw was shredded into 0–2 mm particles, 1 g portions were placed in the small nylon sachets and were subjected to E-beam sterilization. Wetted sachets (10 per container) were placed vertically in rows at a depth of 0.5–4 cm inside seven replicate containers with pre-prepared soil. Additionally, five replicate control containers with soil and without straw were laid at the same time. More detailed information about the experiment layout was described earlier [83]. The humidity of the substrates was kept constant at 60% and the temperature was maintained at 28 ± 1 °C for the duration of the experiment.

### 4.2. Microbial Activity Test by the SR Measurement

To assess microbial activity linked to straw decomposition during the experiment, soil respiration (SR) in the experimental and control containers was measured weekly for 6 months using the conventional alkali absorption method [84]. SR data was processed in Statistica 13, using one-way ANOVA with post hoc Tukey HSD test (TIBCO Software Inc., Palo Alto, CA, USA).

### 4.3. Sample Collecting and Amplicon Sequencing

Coinciding with the SR measurements, sachets with straw were pulled one by one out of five experimental containers once every 1–2 weeks for the first 2 months and after that once every 3–4 weeks. Two experimental containers with straw sachets remained intact for all 6 months for the measurement of unaltered SR. The content of pulled-out sachets and the sample of control chernozem soil were stored in plastic tubes at −20 °C for the subsequent molecular analysis. Three to five replications for each time of sampling (thirty-six samples in total) and six replications for the soil sample were used for the DNA extraction with NucleoSpin^®^ Soil Kit (Macherey-Nagel GmbH & Co. KG, Düren, Germany) as described previously (2019).

For the analysis of taxonomic dynamics of straw colonization, libraries of partial 16S rRNA gene (for bacteria and archaea) and of ITS2 (for fungi) were prepared and sequenced on the Illumina Miseq platform (Illumina, Inc., San Diego, CA, USA) as described previously [85,86].

### 4.4. Amplicon Data Analysis

Data from the sequenced amplicon libraries were processed using the DADA2 pipeline [87] in the R software environment v. 4.2 [88]. Taxonomic identification was carried out using the Silva 138.1 database [89] for 16S rRNA gene sequences and the Unite database [90] for ITS2 sequences. The phylogenetic tree was constructed using the SEPP [91] for 16S rRNA data and IQ-TREE 2.1.2 program [92] for ITS2. Further processing was carried out using the phyloseq [93] and ampvis2 [94] packages. Alpha diversity was assessed by observed, Shannon [95], and inverted Simpson [96] measures and MPD from picante [97], with the significance of mean differences between them calculated using ANOVA with Tukey HSD test [98]. Beta diversity was accessed by NMDS [99] with the Bray–Curtis distance matrix [100]. The significance of differences between microbial communities was estimated by PERMANOVA [101] from the adonis2 test in vegan [102]. The WGCNA [103] method after variance stabilizing transformation from DESeq2 [104] was used to distinguish the microbial association into groups characteristic of different colonization phases.

### 4.5. Full Metagenome Sequencing and GH Gene Analysis

To assess the composition of GH genes in the cellulolytic community the DNA isolated from the 3-month composting straw sample was used for the full metagenome sequencing using the MinION platform (Oxford Nanopore Technologies, Oxford, UK) as described previously [44]. The resulting raw reads were base-called using guppy v. 6.0.6 [105] with a high-accuracy model and clipping adapter sequences and were additionally checked for adapter sequences using porechop v. 0.2.4 [106], which were removed. Flye v 2.9 with a --meta flag [107] was used to assemble the metagenome from the reads. Assembly was polished using a single run of medaka v. 1.5.0-rc.2 [108], which was used for subsequent steps. The assembly was annotated using eggNOG-mapper v. 2.1.9, using -m diamond and --dmnd_frameshift [109]. The search for GH genes was conducted using hmm profiles from the PHAM database [110]. The attribution of GH genes to different functional groups was performed using the CAZy database [8]. 

### 4.6. Real-Time PCR Analysis

To evaluate dynamics of microbial content in the decomposing substrate real-time PCR of 16S rRNA fragment for bacteria and ITS2 fragment for fungi was conducted in triplicate for samples from day 28, 91 and 161 on the CFX96 Real-Time PCR Detection System (Bio-Rad, Germany) as described previously [85]. The threshold cycle (C_T_) data was converted to the number of ribosomal operons per 1 g of substrate. The significance of mean differences between different days of measurement was calculated using ANOVA with Tukey HSD test [98].

To assess the dynamics of GH genes in the duration of the experiment, we constructed primers on the representative set of bacterial GH genes found in the metagenome and belonging to the genera detected by 16S rRNA gene sequencing (in total 23 primer pairs (Appendix A)). The real-time PCR was performed with these primers for samples from days 3, 28, 91 and 161 in triplicate. As an internal control 16S rRNA gene was used. The real-time data was processed using the comparative C_T_ method (the 2^-ΔΔCT^ method) [111] and clusterizied using WPGMA based on Euclidean distance in R. ANOVA with Tukey HSD test was applied on the log transformed values.

The code is available at https://crabron.github.io/manuals/straw_wgcna.html, accessed on 9 December 2022.

## 5. Conclusions

The novelty of this work consisted in the design of the experiment, which demonstrated the dynamics of microbial de novo colonization of straw substrate by soil microbiota during a 6-month period. Chernozem soil acted as a primary source of cellulolytic microorganisms, whose abundance strongly shifted during straw decomposition. The process of bacterial succession was accompanied with the decrease of microbial activity, but the increase in diversity of bacteria during the experiment. However, no increase in diversity was shown for the fungal community. Bacterial succession was divided into three phases, each characterized by a group of co-changing taxa. Genes from various GH families have been detected in the community since the first phase, with the largest increase in the middle phase. The changes of the selected GH genes representation between phases were explained by their taxonomic rather than functional attribution. The early phase was characterized by the appearance of representatives of Bacteroidetes and Alpha- and Gammaproteobacteria, which were shown as potential active decomposers of lignocellulose substrate but which disappear by the end of the phase. The middle phase can be considered the core of the emerging cellulolytic community, most of which contain GH genes, connected with cellulose decomposition, according to the metagenome sequencing, including bacteria (*Chitinophaga*, *Bacillus*, *Ohtaekwangia*, Rhizobiaceae) and fungi (*Chloridium* and Coniochaetaceae). The last phase marked the functional diversification of the community, when predatory microorganisms and bacteria involved in the cycling of other non-carbon substrates and released as a result of the activity of other microorganisms appear. All this may suggest that we should not consider cellulosic communities only as a source of GH-rich microorganisms; a comprehensive approach is required to construct stable and effective decomposing communities.

## Figures and Tables

**Figure 1 ijms-24-06342-f001:**
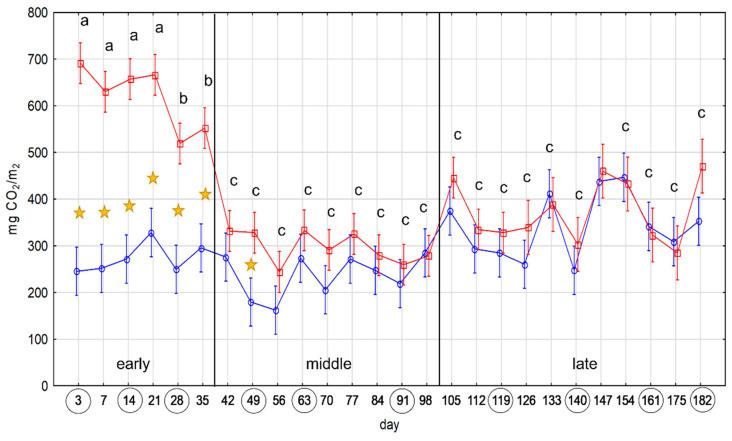
Soil respiration (SR) data. On the Y-axes are SR values in mg CO_2_ per m_2_, and on the X axis are days of measurement. The blue line corresponds to the SR values of containers with soil without added straw (control). The red line corresponds to SR values of containers with soil with added straw in sachets (experiment). Vertical red and blue bars denote 0.95 confidence intervals. Stars mark significant (*p*-value ≤ 0.05) differences between the experiment and control values. Groups of experimental SR values without significant (*p*-value ≤ 0.05) differences are marked with the same letter (‘a’, ‘b’, and ‘c’). Vertical black bars mark different phases of decomposition: early, middle, and late. Circles around days on the *X*-axis denote the samples chosen for amplicon sequencing.

**Figure 2 ijms-24-06342-f002:**
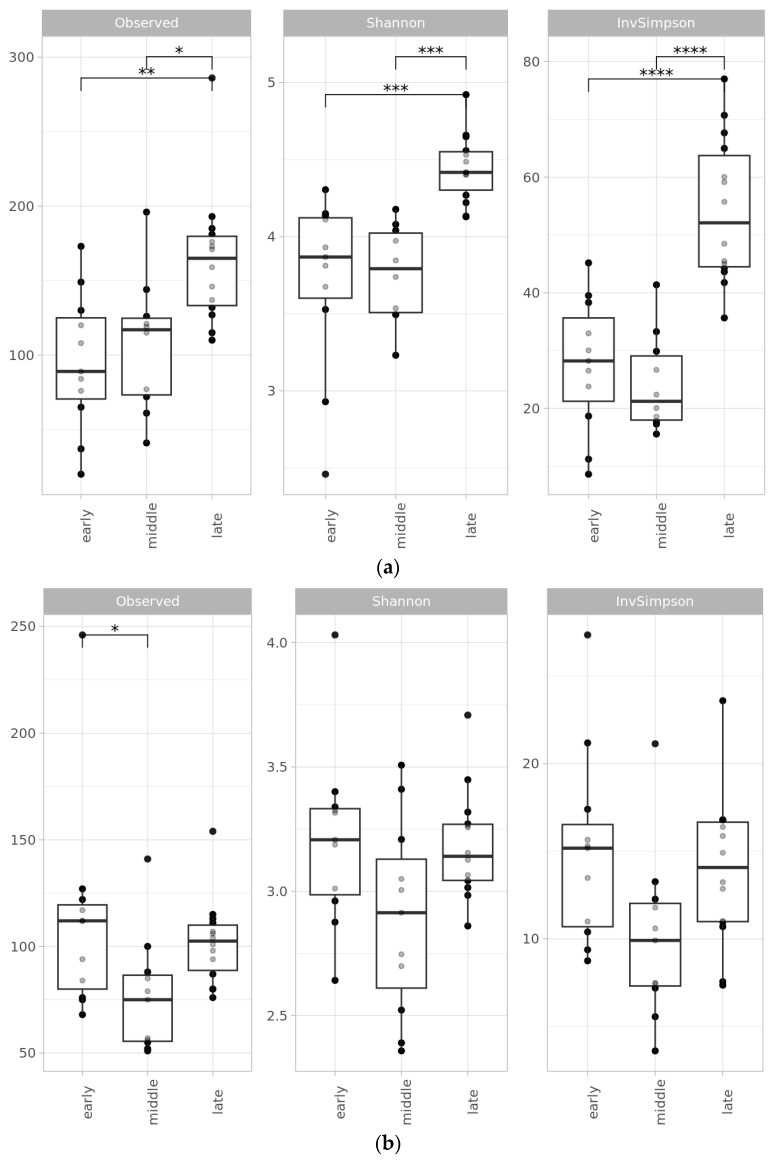
Alpha diversity indices (observed, Shannon, InvSimpson) for (**a**) prokaryotic, based on 16S rRNA gene amplicon libraries, and (**b**) fungal data, based on ITS2 fragment amplicon libraries, divided on the *X*-axis into three phases: early, middle, and late. Significant differences were assessed by ANOVA with Tukey HSD test: (*) *p*-value ≤ 0.05; (**) *p*-value ≤ 0.01; (***) *p*-value ≤ 0.001; (****) *p*-value ≤ 0.0001. (**c**) Alpha-diversity indices of prokaryotic community for each sampling day compared to SR data, expressed in z-scaled values.

**Figure 3 ijms-24-06342-f003:**
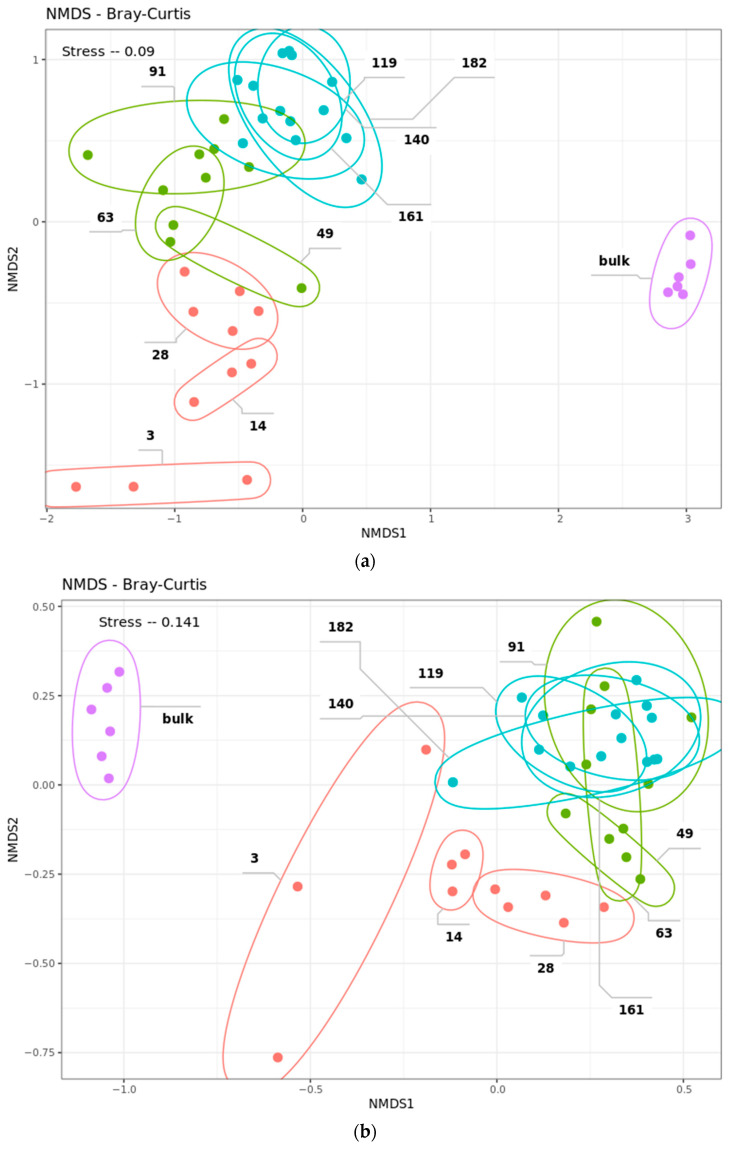
NMDS plot of the beta-diversity of the microbial straw decomposing community from ten sampling points accessed with Bray–Curtis, based on (**a**) 16S rRNA gene and (**b**) ITS2 amplicon sequencing data. Replicates of each sampling day are surrounded by ellipses. The number of the ellipse indicates the day from the beginning of the experiment when the sample was taken. The color represents phase of the experiment: red—early phase, green—middle phase, blue—late phase, purple—control chernozem soil (bulk).

**Figure 4 ijms-24-06342-f004:**
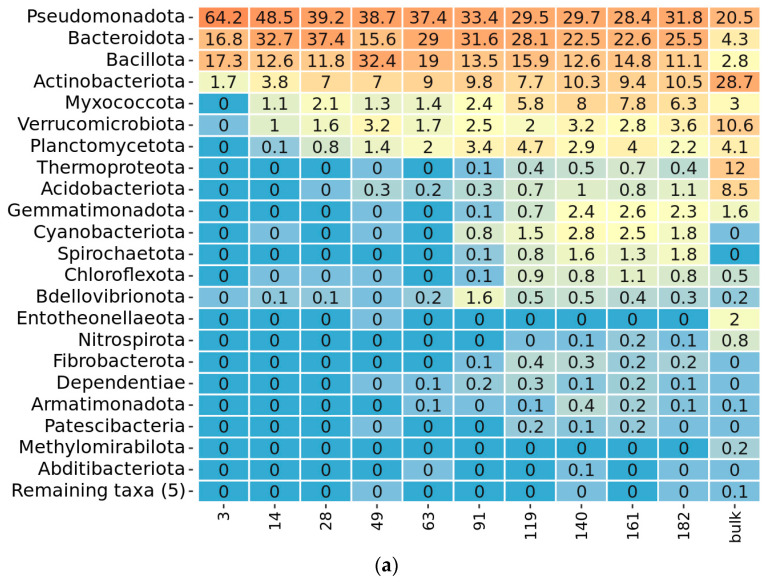
Heatmap of the most abundant phyla in the microbial straw decomposing community from ten sampling points, based on (**a**) prokaryotic and (**b**) eukaryotic amplicon sequencing data. The relative abundance is given in % of the read count of each sampling day, with orange for maximal and blue for minimal values. The name of each sample indicates the day from the beginning of the experiment when the sample was taken. Bulk stands for the control sample of chernozem soil.

**Figure 5 ijms-24-06342-f005:**
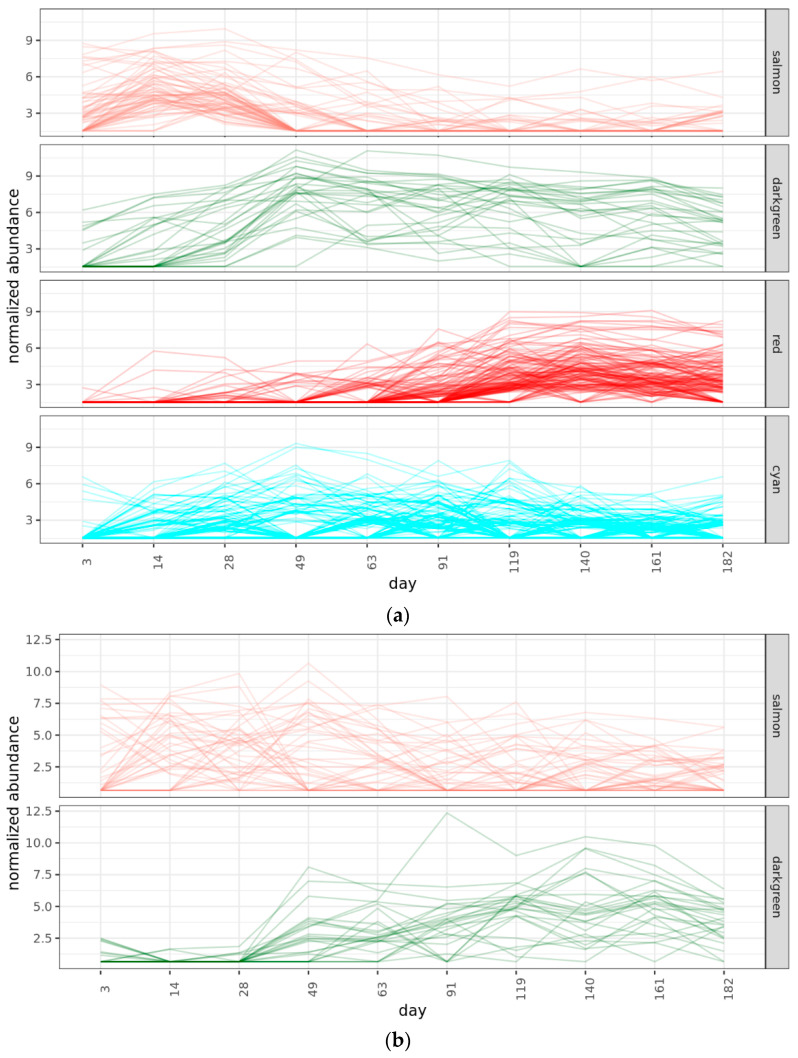
Clusters of co-changing groups of phylotypes for (**a**) prokaryotic and (**b**) fungal amplicon sequencing data, as assessed by WGCNA. (**a**) Early phase—salmon cluster, middle phase—green clyster, late phase—red cluster, universal—cyan cluster. (**b**) Early phase—salmon cluster, late phase—cyan clyster, universal—green cluster.

**Figure 6 ijms-24-06342-f006:**
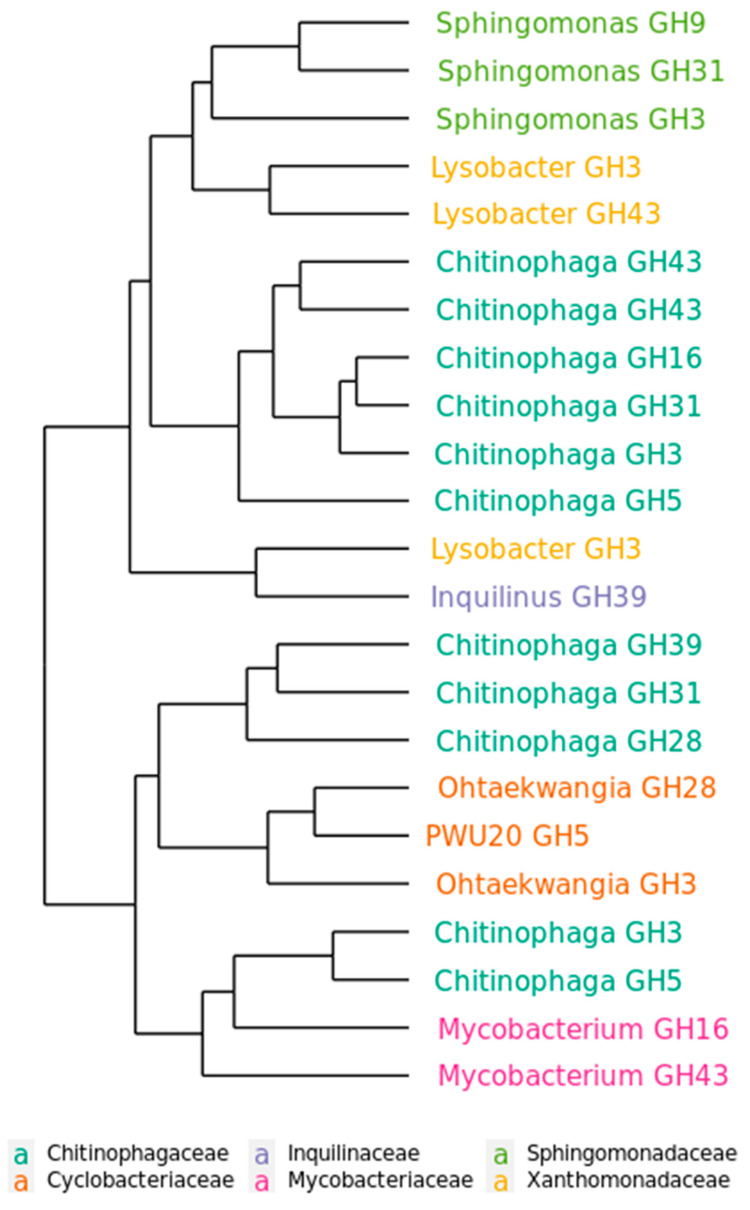
The WPGMA clusterization of the dynamics of selected GH genes across three decomposition phases accessed by the real-time analysis. Each node represents real-time data for one of the GH genes used in the analysis. The name of the node includes taxonomic (on the genus level) and GH family attribution of the gene. The color of the node name represents taxonomic attribution on the family level.

**Table 1 ijms-24-06342-t001:** The distribution of the main GH families found in the metagenome of the cellulolytic community between phyla. Three groups of GH families were distinguished: “cellulose” (GH3, GH5, GH9, GH43, GH94, GH30, etc.), “carbohydrates” (GH31, GH95, GH15, GH77, GH38, GH32, etc.), and “chitin” (GH18, GH19, GH20).

Kingdom	Phylum	“Cellulose” Group	“Carbohydrates” Group	“Chitin” Group
Archaea	Euryarchaeota	0	1	0
Bacteria	Pseudomonadota	239	113	67
	Bacteroidota	126	123	67
	Actinobacteriota	60	52	34
	NA	46	27	14
	Bacillota	19	22	3
	Acidobacteriota	14	13	4
	Planctomycetota	8	11	1
	Verrucomicrobiota	1	8	0
	Cyanobacterota	3	3	0
	Chloroflexota	3	2	0
Fungi	Ascomycota	55	56	68
	Basidiomycota	0	0	1
	NA	0	1	0
	Total	574	432	259

## Data Availability

Data are available at the NCBI Project PRJNA841641, BioSamples: SAMN28605563, SAMN31919897-SAMN31919906.

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
