# Peer review of "The Succession of the Cellulolytic Microbial Community from the Soil during Oat Straw Decomposition"

_ijms, 2023, doi:10.3390/ijms24076342_

Round 1
Reviewer 1 Report
Strong paper; well-described and discussed. Appropriate inferences drawn. Did wish to know more about the history of your chernozem soil. Did think that you could establish your team's qualifications for this sort of research in the introduction.
line 28: Would it be possible to explain this reference at this point "GH genes" for enhanced clarity (as a stand-alone abstract)?
line 37: in a fair number of cases, the harvest index (grain weight in proportion to above-ground total biomass) now exceeds 0.50. Of course, the residue is still an important fraction of the material "essence" of the crop grown. Unless removed, its in situ degradation (by these microbes, primarily) is indeed very important.
line 51: Apart from my concern above, a very effective introduction to this topic!
line 61: Very good points on the limitations of single strain research regarding breakdown of lignins and cellulose.
line 91: You clearly establish the need for succession research in this paragraph.
line 167: I hope that you come back to this divergents in your interpretative section later.
line 171: I guess that I'll find out later (Methods?) what your soil source of innoculum was--perhaps a chernozem soil "experienced" in degrading oat straw?
line 332: Perhaps refer to this work in your introductory materials?
line 344: Good inference about your specific substrate added to your soil.
line 358: Intriguing and important point--might this indicate that this soil included "pre-trained" microbial communities?
line 411: thoughtful explanation of the importance of these diverse bacteria
line 449: Would it be possible to include some "soil history" in this paragraph? What were recent crops, how was residue managed, how long has the soil been under cultivation, have any anti-microbial/anti-fungal pesticides been recently applied?
line 465: Without going to read ref #82, I am unsure about this method. I taught fundamentals of soil science for several years--had good results for inexperienced students working in pairs in lab with diverse soils and different C:N ratios of added plant residue. We used sealed canning jars with known quantities of an alkaline CO2 trapping solution in open beakers inside (then titrated>) Were you doing something like this?
Author Response
Responses to the reviewer's comments are presented in a separate file.

Reviewer 2 Report
This manuscript described on the succession of the microbial community of the soil during straw decomposition.
This research is excellent that the authors analyzed not only the changes in the microbiota from both bacteria and fungi, but also investigates the genes for cellulose degradation by metagenomic analysis. However, this reviewer felt sorry that there is no data showing the relationship between the degree of degradation of straw and the data on microbiota changes. Therefore, most of the spaces are occupied by descriptive contents for the results of microbiome analysis and the like. It is desirable that this manuscript should be revised to easier to understand the straw decomposition process in the three phases shown in this manuscript.
1) If possible, please consider the relationship between physicochemical environmental factors and microbiota changes. Please discuss on the relationship between them.
2) If possible, please show the data for changes in abundance of bacteria and fungi.
3) Please show the data of bacterial and fungal microbiota in control (without straw) and discuss the differences.
4) Figure S2 is important data showing microbiota transition and other measurement results. Therefore, it is recommended to bring Figure S2 from supplementary data to the main body of the paper. Please discuss the relationship between them.
5) In the first paragraph of the Discussion, please discuss the most important findings of this research and please evaluate the important finding. In addition, please discuss what progress has been made by this study in considering of previously reported related studies.
6) Objective of this study is not clear in abstract. In addition, key finding of the study is not clearly articulated in the abstract.
7) It is difficult to understand in which decomposition process (among the three phases) the presence of the GH gene was high.
Author Response

(The authors gave the same response as above.)

Reviewer 3 Report
Interesting article, contains elements of novelty and substantive discussion. Nevertheless, I have a few comments:
1. Extend the abstract, as well as conclusions about the key results for the study. Not all of them have been presented.
2. The results of subchapter 2.1 concerning soil enzymatic activity are too laconic.
3. The chapter on research methodology should appear after the introduction and be divided into clear sections. Have you presented all statistical methods used in the methodology?
4. Unfortunately, your drawings are illegible. I suggest separating the grouped diagrams - in the current form the font is too small.
5. When discussing the results - you should refer to the results and theoretical issues contained in the literature on an ongoing basis, on the basis of which you can interpret the results obtained. Your literature database is already rich - use it.
Author Response

(The authors gave the same response as above.)

Round 2
Reviewer 2 Report
This manuscript described on the succession of the microbial community of the soil during straw decomposition has been revised.
Although I feel sorry that it is difficult to understand the relationship between straw decomposing rate and the transitional phase change. However, the important points became clear by this revision.
This reviewer still has additional comments described below.
1. The advocate for the separation for three phases is very important. Therefore, please define each three phases considering various parameters.
2. It seemed that dramatic difference between the experiment (added straw) and the control (without added straw). Why the authors analyzed functional distribution employed the late middle phase? The reason may describe in L346-348. However, it seems that it is not clear the relationship between decomposing intensity of straw and the diversity of GH genes.
3. There was a pronounced differences between experiment (added straw) and control (without added straw). Please show analyses result form common feature and the difference (e.g., Venn diagram and LEfSe).
4. Please discuss differences in functions in decomposition of straw between prokaryotes and fungi.
5. Names of phylum have been changed recently (e.g., Firmicutes→Bacillota).
6. Please present the data of bacterial quantities assessed by the real-time PCR (L393) as a supplementary material.
Author Response
The answers to the reviewer are attached below.

Round 3
Reviewer 2 Report
The paper described on the succession of the cellulolytic microbial community from the soil during oat straw decomposition has been revised again.
This reviewer noticed the authors sincerely considered the reviewer’s opinion and the manuscript has been much improved compared with the previous version.
1) At the first paragraph of “Discussion” the authors should discuss the most important finding of this study considering of the background studies.
2) The abstract is written descriptively about the results obtained, and it is difficult to understand the advanced points of this study considering of previous studies.
3) Conclusion (L654-655): Although the authors mentioned that “quantity”, the quantitative approach was not presented as Table or figure. The quantitative data should be used in the main body manuscript.  
4) If possible, please describe author’s views on the differences in the contribution of bacteria and fungi to straw decomposition and their interactions in the Discussion.
5) If possible, please explain what happened in the straw decomposition or microbial community in each phase in the three phases in summary. It is difficult to understand the differences between 1st and 2nd phases from the points what have happened for the degradation of oat strew by reading Abstract.
Author Response
We are very glad to be able to improve our manuscript according to your valuable suggestions.
Below in italics are the answers to the reviewers’ comments.
The paper described on the succession of the cellulolytic microbial community from the soil during oat straw decomposition has been revised again.
This reviewer noticed the authors sincerely considered the reviewer’s opinion and the manuscript has been much improved compared with the previous version.
1) At the first paragraph of “Discussion” the authors should discuss the most important finding of this study considering of the background studies.
Thank you for your patience! I totally forgot among all other comments to address this proposition in the first round of the review. Our design experiment is similar to previously published papers, but ours shows true colonization of a sterilized lignocellulose substrate from the external source of the microbiota. This fact is pointed out in the abstract and conclusions. The first paragraph of the discussion was also modified.
2) The abstract is written descriptively about the results obtained, and it is difficult to understand the advanced points of this study considering of previous studies.
The abstract was corrected to more clearly present findings about succession and isolated phases of decomposition.
3) Conclusion (L654-655): Although the authors mentioned that “quantity”, the quantitative approach was not presented as Table or figure. The quantitative data should be used in the main body manuscript.  
The mention of the “quantity” was removed from the conclusions. But the data itself remains in the manuscript, presented in the Figure S1.
4) If possible, please describe author’s views on the differences in the contribution of bacteria and fungi to straw decomposition and their interactions in the Discussion.
Last paragraph of the discussion was modified to briefly mention this issue.
5) If possible, please explain what happened in the straw decomposition or microbial community in each phase in the three phases in summary. It is difficult to understand the differences between 1st and 2nd phases from the points what have happened for the degradation of oat strew by reading Abstract.
Clarification was added to the abstract. First and second phases are quite close, but the difference is that the first phase is characteristic of high microbial activity and low diversity, while the middle phase has low activity and low diversity. It can be considered as an “intermediate” phase. The late phase has low activity, but bacterial diversity is high.